# Linear and Kernel Classification in the Streaming Model: Improved Bounds for Heavy Hitters

**Arvind Mahankali**
CMU
amahanka@andrew.cmu.edu

**David P. Woodruff**
CMU
dwoodruf@cs.cmu.edu

## Abstract

We study linear and kernel classification in the streaming model. For linear classification, we improve upon the algorithm of [1], which solves the $\ell_1$ point query problem on the optimal weight vector $w_* \in \mathbb{R}^d$ in sublinear space. We first give an algorithm solving the more difficult $\ell_2$ point query problem on $w_*$, also in sublinear space. We then give an algorithm which solves the related $\ell_2$ heavy hitter problem on $w_*$, in sublinear space and running time. Finally, we give an algorithm which can *deterministically* solve the $\ell_1$ point query problem on $w_*$, with sublinear space, improving upon that of [1]. For kernel classification, if $w_* \in \mathbb{R}^{d^p}$ is the optimal weight vector classifying points in the stream according to their $p^{th}$-degree polynomial kernel, then we give an algorithm solving the $\ell_2$ point query problem on $w_*$ in $\text{poly}(\frac{p \log d}{\varepsilon})$ space, and an algorithm solving the $\ell_2$ heavy hitter problem in $\text{poly}(\frac{p \log d}{\varepsilon})$ space and running time. Note that our space and running time is polynomial in $p$, making our algorithms well-suited to high-degree polynomial kernels and the Gaussian kernel (approximated by the polynomial kernel of degree $p = \Theta(\log T)$). Our algorithms for kernels are in fact a special case of a more general algorithm we give for low-rank tensor inputs.

## 1 Introduction

We consider logistic regression, and more generally, linear classification, in the streaming model. In our setting, we are given a dataset consisting of $T$ examples $(x_t, y_t)$, where $t \in [T]$, $x_t \in \mathbb{R}^d$, $y_t \in \{-1, 1\}$. The examples arrive one by one, and moreover, the nonzero coordinates of each example $x_t$ arrive one by one. Our goal is to estimate the weights $w_* \in \mathbb{R}^d$ of the optimal linear classifier for these examples. Here, $w_* := \operatorname{argmin}_{w \in \mathbb{R}^d} \frac{1}{T} \sum_{t=1}^{T} \ell(y_t w^T x_t) + \frac{\lambda}{2} \|w\|_2^2$ where $\ell$ is a loss function satisfying certain conditions described in Section 1.3 — the prototypical example of $\ell$ that we consider is the logistic regression loss function — and $\lambda$ controls the strength of the $\ell_2$ regularization. Finally, we assume that $d$ is very large, and we therefore wish to estimate the weights of $w_*$ in space that is sublinear in $d$. This is important both in settings with devices with limited memory constraints, such as routers or sensors on a network, as well as in machine learning problems with many features. Machine learning problems with a very large number of features arise in many natural language processing tasks, for example — one motivation for [1], the precursor to our work, is that the use of $n$-gram features when analyzing text data can lead to a very large memory cost. [1] Our goal, and that of [1], is to find features which are the most useful for classification — as pointed out in [1], previously known sketches for compressing classifiers do not achieve this goal.

Formally, we consider the following problems in this work:

---

[1]In [1] it is mentioned that in an experiment on the dataset of [2], "we recorded 47M unique word pairs that co-occur within 5-word spans of text ... [requiring] approximately 560MB of memory."

35th Conference on Neural Information Processing Systems (NeurIPS 2021).

**Problem 1** ($\ell_p$ Point Query, for $p = 1, 2$). *Let $\varepsilon \in (0, 1)$. At any time $t \in [T]$ in the stream, given an arbitrary query $i \in [d]$, the goal is to output $\widehat{w_i}$ such that $|\widehat{w_i} - w_{*,i}| \leq \varepsilon \|w_*\|_p$.*

**Problem 2** ($\ell_2$ Heavy Hitters). *Let $\varepsilon \in (0, 1)$. At any time $t \in [T]$ in the stream, the goal is to output a list $L \subset [d]$ of size at most $O(\frac{1}{\varepsilon^2})$ such that $L$ contains all indices $i \in [d]$ such that $|w_{*,i}| \geq \varepsilon \|w_*\|_2$.*

Interpretability is one of the main motivations for the above problem formulations — as argued by [1], finding the largest weights in $w_*$ is equivalent to determining which features are the most important in classification. Also, note that $\ell_2$ point query is strictly more difficult than $\ell_1$ point query — to see why, note that in the worst case, $\|w_*\|_1$ can be larger than $\|w_*\|_2$ by a $\sqrt{d}$ factor. Thus, for instance, if an algorithm uses $\text{poly}(\varepsilon^{-1} \log d)$ space to solve $\ell_1$ point query, then it will use at least $d^{O(1)}$ space to solve $\ell_2$ point query (by replacing $\epsilon$ with $\epsilon/\sqrt{d}$), which is far too large in streaming settings.

## 1.1 Our Contributions

**$\ell_2$ point query and heavy hitters (via a reduction to turnstile $\ell_2$ point query and heavy hitters)**
We give efficient algorithms for solving $\ell_2$ point query and $\ell_2$ heavy hitters. In the approach of [1], a single Countsketch matrix [3] is used to maintain a sketch $z_t$ of the weights. $z_t$ is updated by online gradient descent according to $z_{t+1} \leftarrow (1 - \lambda \eta_t) z_t - \eta_t y_t \ell'(y_t z_t^T R x_t) R x_t$, where $R$ is a Countsketch matrix scaled by $1/\sqrt{s}$, where $s$ is the column sparsity of $R$ — by [4], $R$ is a Johnson-Lindenstrauss (JL) transform if $s$ is chosen to be large enough. To estimate the coordinates of $w_*$, the recovery procedure of [3] is applied to $\sqrt{s}\overline{z}$, where $\overline{z} = (\sum_{t=1}^T z_t)/T$. In addition, [1] only obtains the $\ell_1$ point query guarantee, since the JL property of $R$ is only applied to show that $R$ preserves the inner products between $e_1, e_2, \ldots, e_d$.

To resolve both of these issues, we decouple the JL matrix from the point query/heavy hitters sketch, i.e., we use a JL matrix $R$, and a separate sketch $S$ for $\ell_2$ point query/heavy hitters in the well-studied turnstile streaming model. First, we maintain $z_t$, which is updated using online gradient descent as in [1]. In addition, we maintain an additional vector $\widehat{w_t} \in \mathbb{R}^d$, which is updated according to $\widehat{w_{t+1}} \leftarrow (1 - \lambda \eta_t) \widehat{w_t} - \eta_t y_t \ell'(y_t z_t^T R x_t) x_t$. The motivation for this update is that it is essentially the online gradient descent update $w_{t+1} \leftarrow (1 - \lambda \eta_t) w_t - \eta_t y_t \ell'(y_t w_t^T x_t) x_t$ we would perform without any sketching, but we replace $\ell'(y_t w_t^T x_t)$ with $\ell'(y_t z_t^T R x_t)$ due to space constraints. We do not have enough space to explicitly maintain $\widehat{w_t}$, but since the updates to $\widehat{w_t}$ are additive, we can still give it as the input to a turnstile streaming algorithm for $\ell_2$ point query/heavy hitters. We show that $\overline{\widehat{w_T}} = \frac{1}{T} \sum_{t=1}^T \widehat{w_t}$ is close to $w_*$ in $\ell_2$ norm, and thus it suffices to solve $\ell_2$ point query and $\ell_2$ heavy hitters on $\overline{\widehat{w_T}}$. In summary, we give algorithms for $\ell_2$ point query and heavy hitters with $O(\varepsilon^{-2} \log(dT/\delta))$ space and $1 - \delta$ success probability.

**Deterministic $\ell_1$ point query** In addition, we show that $\ell_1$ point query can be solved *deterministically*, and with space complexity $O(\varepsilon^{-2} \log(d))$, which is smaller than the $O(\varepsilon^{-4} \log^3(d/\delta))$ space complexity of [1]. Deterministic sketches are useful as they allow for inputs to be chosen as a function of past responses of the sketching algorithm, and thus provide adversarial robustness [5]. To obtain a deterministic algorithm, we replace the Countsketch matrix used by [1] with an $\varepsilon$-incoherent matrix. Here, an $\varepsilon$-incoherent matrix $R \in \mathbb{R}^{s \times d}$ is one whose columns are *almost orthonormal*, meaning that for all $i \neq j$, $|\langle R_i, R_j \rangle| \leq \varepsilon$, and all columns of $R$ have $\ell_2$ norm 1. Matrices that are $\varepsilon$-incoherent were previously applied to streaming problems by [6], and can be constructed deterministically. To improve on the space complexity of [1], we change the recovery procedure: to estimate $w_{*,i}$, we simply compute $\langle R_i, z \rangle$ where $z$ is the compressed weight vector, rather than applying the Countsketch recovery procedure of [3].

**$\ell_2$ point query (via a combined JL/point query sketch)** Inspired by our deterministic $\ell_1$ point query algorithm, we provide an alternative algorithm for $\ell_2$ point query, for which the space complexity has a smaller dependence on $1/\lambda$. We observe that the sparse JL transform of [4] can be used not only to preserve norms with high probability (i.e., to satisfy the JL lemma) but also to provide an $\ell_2$ point query sketch directly, using a different $\ell_2$ point query recovery procedure. Our procedure does not involve any median based operations. Instead, to estimate $w_{*,i}$ given the compressed weight vector $z$, we simply compute $\langle R_i, z \rangle$. Recall that [1] multiplies the sketching matrix $R$ by $\sqrt{s}$ in order to perform the recovery procedure of [3] — our new procedure also avoids this rescaling, and thus achieves a space complexity of $O(\varepsilon^{-2} \log(dT/\delta))$ up to problem-dependent parameters. The space complexity of this algorithm has a smaller dependence on $1/\lambda$ compared to our $\ell_2$ point query

algorithm discussed above. In addition, we use online gradient descent regret bounds to show that the estimation error of our algorithms involves a term that is proportional to $\frac{1}{T^{1/4}}$. Thus, for the error to be at most $\varepsilon \|w_*\|_2$ or $\varepsilon \|w_*\|_1$, $T$ must be at least a certain value. Our $\ell_2$ point query algorithm using a combined JL/point query sketch requires $T$ to grow as $\frac{1}{\lambda^4}$, as opposed to our $\ell_2$ point query algorithm which makes a black-box reduction to the turnstile model, which requires $T$ to grow as $\frac{1}{\lambda^8}$.

The main idea of this algorithm is the observation that if $w_*$ is the optimal weight vector, then $\langle Rw_*, Re_i \rangle$ is a good estimate of $\langle w_*, e_i \rangle = w_{*,i}$ — this motivates our query procedure. Note that this fact is implicit in the guarantees of a JL matrix. This recovery procedure has also been used for turnstile $\ell_1$ point query by [6] (which motivated our deterministic $\ell_1$ point query algorithm above). The same recovery procedure was also used by [7], in the context of distributed differentially private heavy hitters. To our knowledge, our work is the first to use this idea in the setting of $\ell_2$ point query for linear classification. Note that for $\ell_2$ point query in the turnstile model, it is preferable to use Countsketch (Countsketch is also used by [1]), since for an update to a single coordinate, the update time with Countsketch is $O(\log(1/\delta))$, while the update time when using a sparse JL matrix [4] is $O(\varepsilon^{-1} \log(1/\delta))$. However, in the context of linear classification, we find that using a JL matrix with the recovery procedure $\langle Rw_*, Re_i \rangle \approx w_{*,i}$ reduces the space complexity by a factor of $\text{poly}(\varepsilon^{-1} \log(d/\delta))$, as long as $T = O(d)$. This is because the sketching matrix already needs to be a JL matrix in order to preserve certain inner products. In this case, using the Countsketch recovery procedure requires scaling $\overline{z}$ by a factor of $\sqrt{s}$ where $s$ is the column sparsity of $R$, which in turn requires increasing the accuracy parameter $\varepsilon$ of $R$ in order to solve $\ell_2$ point query (or $\ell_1$ point query).

**Worst-case data order guarantees** For all of our algorithms, we do not make assumptions on the order of the $x_t$, unlike [1]. In [1] the pairs in the set $\{(x_1, y_1), \ldots, (x_T, y_T)\}$ are required to arrive in the stream in a uniformly random order. The following is given in [1] as a heuristic explanation: "we believe this condition is necessary to avoid worst-case adversarial orderings of the data points - since the WM-Sketch update at any time step depends on the state of the sketch itself, adversarial orderings can potentially lead to high error accumulation ... Intuitively, it seems reasonable to expect that we would need an 'average case' ordering of the stream in order to obtain a similar recovery guarantee to the batch setting." It is perhaps surprising then that we are able to entirely remove this assumption. We do this by showing that instead of using Corollary 1 of [8] (which is used by [1]) we can use an argument from first principles based on online gradient descent regret bounds.

**Classification with tensor inputs** We consider a variant of linear classification where the inputs $x_t$ and the weight vector $w_*$ are $p$-th order tensors (i.e., are vectors in $\mathbb{R}^{d^p}$) and moreover, the $x_t$ have rank at most $k$, meaning $x_t = \sum_{i=1}^{k} x_t^{(i,1)} \otimes x_t^{(i,2)} \otimes \ldots \otimes x_t^{(i,p)}$, where the $x_t^{(i,j)} \in \mathbb{R}^d$. This is motivated by applications of tensor regression, for instance in neuroimaging [9, 10], where the covariates have a tensor product structure. Furthermore, the $x_t$ may be of low rank in applications — for instance, in the case $p = 2$, [9] mentions that in [10], tensor regression is performed after principal component analysis is first performed on the $x_t$. In such a setting, we wish to obtain $\ell_2$ point query and heavy hitters algorithms with at most a polynomial dependence on $\log d$ and $1/\varepsilon$, and moreover a polynomial dependence on $p$. To achieve this, we use tensor sketching techniques of [11], which develops a sketching matrix $M \in \mathbb{R}^{m \times d^p}$, where $m = \text{poly}(\varepsilon^{-1} p \log d)$, such that $M$ is a JL matrix, and $Mx^{\otimes p}$ can be computed very efficiently for $x \in \mathbb{R}^d$ (specifically, in $\text{poly}(\varepsilon^{-1} p \log d) \cdot \text{nnz}(x)$ time), without explicitly forming $x^{\otimes p}$. Thus, for $\ell_2$ point query, we can use $M$ in the same way we use the sparse JL matrix of [4] in the combined JL/point query sketch above.

For $\ell_2$ heavy hitters, our algorithm is as follows: (1) for each mode $i \in [p]$, we determine the coordinates $j \in [d]$ which contribute more than an $\varepsilon$ fraction of the $\ell_2$ norm of $w_*$ — in other words, we want to find all $j \in [d]$ such that $\|w_*(:, \ldots, :, j, :, \ldots, :)\|_2 \geq \varepsilon \|w_*\|_2$, where $w_*(:, \ldots, :, j, :, \ldots, :)$ consists of those coordinates of $w_*$ which have index $j$ in the $i^{th}$ mode. This gives us a list $L_i \subset [d]$ of size at most $O(1/\varepsilon^2)$, for each $i \in [p]$. (2) Then, we find the (at most $O(1/\varepsilon^2)$) indices $(i_1, \ldots, i_p)$ of $w_*$ in $[d]^p$ such that $|w_*(i_1, i_2, \ldots, i_p)| \geq \varepsilon \|w_*\|_2$. We do step (2) using the $L_i$, by inductively constructing *prefixes* of these coordinates, one mode at a time. For each $i \in [p]$, we build an auxiliary data structure which can estimate $\|w(j_1, \ldots, j_i, :, \ldots, :)\|_2$ for any prefix $(j_1, \ldots, j_i)$ of length $i$ — this is also done by using the sketching matrix of [11]. Both our $\ell_2$ point query and $\ell_2$ heavy hitters algorithms for $p^{th}$-order tensor inputs have $\text{poly}(\varepsilon^{-1} p \log(dT/\delta))$ space and query time, and $\text{poly}(\varepsilon^{-1} p \log(dT/\delta)) \sum \text{nnz}(x_t^{(i,j)})$ update time, up to problem-dependent parameters.

When the inputs are $p^{th}$ order tensors of low rank, our $\ell_2$ point query and heavy hitters algorithms for tensor inputs give significant savings in update time when compared to standard $\ell_2$ point query/heavy hitters algorithms. To see why, note that when the $x_t$ are rank-$k$ tensors, the update to $\widehat{w_t}$ (defined above) is $\widehat{w_{t+1}} \leftarrow (1 - \lambda\eta_t)\widehat{w_t} - \eta_t y_t \ell'(y_t z_t^T M x_t) \sum_{i=1}^{k} x_t^{(i,1)} \otimes x_t^{(i,2)} \otimes \ldots \otimes x_t^{(i,p)}$. Using a standard $\ell_2$ heavy hitters algorithm on $\widehat{w_t}$ requires explicitly forming $x_t^{(i,1)} \otimes x_t^{(i,2)} \otimes \ldots \otimes x_t^{(i,p)}$ — if the $x_t^{(i,j)}$ are dense, then standard $\ell_2$ heavy hitters algorithms would require at least $d^p$ update time, as opposed to our algorithm, which only has $\text{poly}(\varepsilon^{-1} p \log(dT/\delta)) \cdot kd$ update time — even when $p = 2$, if $k$ is small, then this is a significant improvement.

**Kernel classification** Kernel logistic regression (KLR) is a well-known classification method in the field of statistical learning, see e.g., [12] and its many citations. We obtain the first results for finding the large weights of a classifier in the kernel space for the polynomial and Gaussian kernels. A succinct summary of the classifier, such as its list of heavy hitters with their approximate values, is especially important for kernel classification, since the dimension of the kernel space can be much larger than $d$, and in the case of the Gaussian kernel, even infinite. In this setting, for the polynomial kernel, classification is done using $x_t^{\otimes p}$ to predict $y_t$ — thus, this is a special case of the setting where $x_t$ is a tensor of rank at most $k$, discussed above. As in [11], we can approximate the Gaussian kernel via a Taylor expansion, using a polynomial kernel of degree $O(\log T)$. Note that if $(i_1, i_2, \ldots, i_p)$ is an index in $[d]^p$ and $(j_1, j_2, \ldots, j_p)$ is a re-ordering of $(i_1, i_2, \ldots, i_p)$, then one may want to consider $x_{i_1} \ldots x_{i_p}$ and $x_{j_1} \ldots x_{j_p}$ to be the same feature. To get around this, suppose $(i_1, i_2, \ldots, i_p)$ has a Hamming distance of at most $c$ from the set of indices of the form $(i, i, \ldots, i)$ and it is an $\varepsilon$-heavy hitter when ignoring permutations (formally defined in the appendix). Then, if we apply our algorithms from the low-rank tensor setting with an accuracy of $\varepsilon/p^{c/2}$, $(i_1, i_2, \ldots, i_p)$ will be detected as a heavy hitter. An interesting open question is whether this can be done in $\text{poly}(\varepsilon^{-1} p \log(dT/\delta))$ space even when $c$ is equal to $p$. We leave this question to future work.

**Experiments** We empirically compare both of our algorithms for $\ell_2$ point query with the WM-Sketch algorithm of [1], [2] where all three algorithms are restricted to certain memory budgets, following the setup of [1]. Our $\ell_2$ point query algorithm that makes use of a combined JL/point query sketch leads to improved performance in estimating $w_*$ compared to the WM-Sketch algorithm, with significantly improved performance for a larger memory budget on the RCV1 dataset [13], though the WM-Sketch algorithm performed better than our black-box reduction-based $\ell_2$ point query algorithm. For smaller memory budgets, these two algorithms appeared to have similar weight recovery performance on the RCV1 dataset, but our other $\ell_2$ point query algorithm using a black-box reduction to turnstile $\ell_2$ point query had much lower error in recovering the top weights.

**Using the Top Weights or Compressed Classifiers for Classification** Here we give additional motivation for estimating the top weights of $w_*$, or applying sketching to classifiers. We performed an experiment on the RCV1 dataset [13], which we divided into a training and testing half — we obtained a weight vector $w \in \mathbb{R}^d$ by using online logistic regression on the training half, and computed the accuracy when using $w^K$ for linear classification on the testing half (where $w^K$ is the $K$-sparse vector whose entries are the top $K$ entries of $w$). One noteworthy result of this experiment is that when $K = 400$, the accuracy on the testing half is 93.9%, while the full weight vector $w$ (which has 41130 nonzero coordinates) achieves 95.7% accuracy. The full details of these experiments are given in Appendix F. We do acknowledge that there are no theoretical guarantees for using only the top $K$ weights for $K \ll d$, and there may be datasets where using the top $K$ weights of $w_*$ may not lead to good performance unless $K$ is very large. We give theoretical guarantees for using a *compressed* classifier, that is, using $\overline{z}^T R$ instead of $w_*$ where $R$ is a sparse JL matrix and $\overline{z}$ is the average iterate of sketched online logistic regression: if $L = \frac{1}{T} \sum_{t=1}^{T} \ell(y_t w_*^T x_t)$ and $\widehat{L} = \frac{1}{T} \sum_{t=1}^{T} \ell(y_t \overline{z}^T R x_t)$, then $|L_* - \widehat{L}| \leq \varepsilon \|w_*\|_2$ as long as $R$ has $O(\varepsilon^{-2} H^2 \log(dT/\delta))$ rows (up to problem dependent parameters) and $T$ is a certain value. We give full details in Appendix E. Finally, we note that in a stream, finding $\ell_2$ heavy hitters in the turnstile model requires $\min(\sqrt{d}/\varepsilon, \log(1/\delta)/\varepsilon^2)$ space, by Theorem 4.3 of [14]. In particular, estimating all the coordinates would require $\text{poly}(d)$ space, meaning that if we wish to obtain sublinear space complexity in our setting, it is reasonable to expect that we cannot do better than estimating the heavy hitters, without additional assumptions.

---

[2]We use the implementation by the authors of [1] at `https://github.com/stanford-futuredata/wmsketch`. Our implementations of our $\ell_2$ point query algorithms are also based on their code.

## 1.2 Related work

**Turnstile Point Query and Heavy Hitters** There is a large body of work on finding the heavy hitters in a data stream. For a survey, see, e.g., [15]. Of particular relevance to this work is the CountSketch algorithm of [3] for finding $\ell_2$ heavy hitters. We note that [16, 17] improve the memory of the algorithm of [3] by a logarithmic factor, but do not handle negative updates, which may arise in our setting. We also need deterministic algorithms for finding $\ell_1$ heavy hitters, and we use the algorithms of [6] which use $\epsilon$-incoherent matrices, and improve upon the earlier work of Ganguly [18]. We note that the CountMin algorithm of [19] also achieves the $\ell_1$ heavy hitter guarantee, though it is randomized, while here we seek a stronger deterministic guarantee. Indeed, for randomized algorithms, we can achieve the stronger $\ell_2$ heavy hitter guarantee.

**Point Query and Heavy Hitters for Classification/Regression** The work that is most closely related to ours is [1], which solves $\ell_1$ point query on $w_*$ in the streaming model, and achieves $O(\varepsilon^{-4} \log^3(d/\delta))$ space up to problem-dependent parameters. Unlike [1], we give an algorithm with provable guarantees for finding the at most $\varepsilon^{-2}$ heavy hitters in *sublinear* time, and we solve the stronger $\ell_2$ point query and heavy hitters problems in addition to $\ell_1$ point query.

Another related work with a somewhat different focus from ours is MISSION [20]. The MISSION algorithm finds a $k$-sparse solution for least-squares regression in low space, using Countsketch. [20] modifies the SGD algorithm: in each iteration, a uniformly random training example is selected and the SGD update is given to a Countsketch data structure — then, Countsketch is used to select the top $k$ features, and the vector with these $k$ non-zero coordinates is used for the next SGD update. MISSION focuses on convergence of the iterates $\beta^t$ to a $k$-sparse vector $\beta^*$, while in the analyses of our $\ell_2$ point query/heavy hitters algorithms, we desire/show convergence in $\ell_2$ norm of $\widehat{w_T}$ to $w_*$ up to additive error $\varepsilon\|w_*\|_2$ for a potentially dense $w_*$. Our gradient updates are thus different, as we perform the update $\widehat{w_{t+1}} \leftarrow (1 - \lambda\eta_t)\widehat{w_t} - \eta_t\ell'(y_t z_t^T R x_t)x_t$, i.e. our update does not involve truncation by taking the top $k$ estimates from Countsketch. We also note that [20] gives a theoretical analysis in the setting where the $x_t$ have i.i.d. Gaussian entries, and $y_t = x_t^T\beta^* + w$, where $w$ is Gaussian noise and $\beta^*$ is a $k$-sparse vector, while we do not assume the inputs/noise are Gaussian.

One more work [21] proposes the BEAR algorithm, which is a sketched version of the online L-BFGS algorithm. The setting of [21] is similar to ours — this work aims to estimate the top coordinates of the weight vector and achieve the $\ell_2$ point query guarantee. The proof of Lemma 3 in [21], that BEAR minimizes a "sketched" version of the loss function, appears to rely on the claim that MISSION minimizes this sketched loss function — however, as noted above, MISSION instead aims to find an optimal $k$-sparse weight vector. We also note that [21] does not propose/analyze an algorithm for recovering all the heavy hitters for the optimal weight vector in sublinear time, while we show how this can be done using turnstile $\ell_2$ heavy hitters algorithms with a small overhead in space/update time and no overhead in query time — fast query time can be useful when the $x_t$, and therefore the gradient updates, are sparse.

**Other Works on Sketching for Classification** We also note that there are a number of other (less closely related) works which use sketching for linear classification — we compare to these works in Appendix A.

To our knowledge, our work is the first to consider linear classification with tensor inputs, and kernel classification, in the streaming model, with the goal of recovering the top weights of these classifiers.

## 1.3 Preliminaries

As in [1], we are given a loss function $\ell$, training examples $\{(x_t, y_t)\}_{t\in[T]}$, $\lambda > 0$, and $w_* := \operatorname{argmin}_{w\in\mathbb{R}^d} \frac{1}{T}\sum_{t=1}^T \ell(y_t w^T x_t) + \frac{\lambda}{2}\|w\|_2^2$. We use online gradient descent regret bounds, specifically Theorem 3.1 of [22]. We use a learning rate of $\eta_t = D/(G\sqrt{t})$, where $D$ is an upper bound on $\|w_*\|_2$ and $G$ is an upper bound on the norm of the gradient ($D$ and $G$ are discussed further in the supplementary material). The following assumptions hold throughout the paper:

**Definition 1.1** (Running Assumptions). *(1) $\ell$ is convex, $\beta$-smooth, and $H$-Lipschitz. (2) For all $t \in [T]$, $\|x_t\|_2 \leq 1$. (3) There exists a constant $\tau > 0$ independent of $T$ such that $\|w_*\|_2 \geq \tau$.*

The last assumption above is not explicitly made in [1], but in Theorem 2 of [1], one of the hypotheses is that $T$ is at least $1/\|w_*\|_1^2$ (omitting other dependencies). Since $w_*$ itself may depend on

$x_1, x_2, \ldots, x_T$, and thus $T$, we explicitly make this assumption to prevent circularity. This assumption is necessary for making use of online gradient descent regret bounds, to ensure $\|w_*\|_2$ does not decrease faster than the average regret.

## 2  $\ell_2$ Point Query and Heavy Hitters

First, we define a "sketched" loss function $\widehat{L}(z) = \frac{1}{T} \sum_{t=1}^{T} \ell(y_t z^T R x_t) + \frac{\lambda}{2} \|z\|_2^2$ as in [1]. Here, $R$ is a sparse Johnson-Lindenstrauss matrix — let us recall the properties of sparse JL matrices:

**Theorem 2.1** (Sparse JL Matrices [4]). *Let $d \in \mathbb{N}$, and $\varepsilon, \delta \in (0, 1)$. Then, there exists a distribution on matrices $S \in \mathbb{R}^{k \times d}$, where $k = \Theta(\varepsilon^{-2} \log(1/\delta))$ and $S$ has $s = \Theta(\varepsilon^{-1} \log(1/\delta))$ nonzero entries per column, such that for any $x \in \mathbb{R}^d$, with probability $1 - \delta$, $(1 - \varepsilon)\|x\|_2 \le \|Sx\|_2 \le (1 + \varepsilon)\|x\|_2$.*

If $z_* := \operatorname{argmin} \widehat{L}(z)$, then $z_*$ is a compressed version of $w_*$ in the following sense:

**Theorem 2.2** (Batch Setting). *Let $\varepsilon, \delta \in (0, 1)$, and suppose $\ell$ is $\beta$-smooth and $\|x_t\|_2 \le 1$ for all $t$. Define $w_* = \operatorname{argmin} L(w)$ and $z_* = \operatorname{argmin} \widehat{L}(z)$. If $R$ is a sparse JL matrix with $O(\varepsilon^{-2} \log(dT/\delta) \cdot \beta/\lambda)$ rows, then with probability $1 - \delta$ over $R$, $\|z_* - Rw_*\|_2 \le \varepsilon \|w_*\|_2$.*

The proof of Theorem 2.2 is in the appendix. It uses a primal-dual argument that is similar to the one used to prove Theorem 1 of [1]. The main difference is that [1] only shows that $\|z_* - Rw_*\|_2 \le \varepsilon \|w_*\|_1$. This is because they only apply the JL property of $R$ to approximately preserve inner products between the vectors $e_1, e_2, \ldots, e_d$. To prove Theorem 2.2, we modify their analysis by also using the JL property of $R$ to additionally show that the inner products $\langle x_t, w_* \rangle$ and $\langle e_i, w_* \rangle$ are well-approximated by $\langle Rx_t, Rw_* \rangle$ and $\langle Re_i, Rw_* \rangle$ respectively, for all $i \in [d]$, and $t \in [T]$.

Our algorithm proceeds by a reduction to standard $\ell_2$ point query and heavy hitters, in the turnstile streaming model. Let us recall the definition of the turnstile streaming model, and the best known results for these two problems in the turnstile model.

**Definition 2.3** (Turnstile Streaming Model). *In the turnstile streaming model, the input is a vector $v \in \mathbb{R}^n$. Updates are of the form $(i, \Delta)$ where $i \in [n]$ and $\Delta \in \mathbb{R}$, signifying that $v_i$ is incremented by $\Delta$. For $\ell_2$ point query in the turnstile model, queries $i \in [d]$ should be answered with an estimate $\widehat{v_i}$ of $v_i$ such that $|\widehat{v_i} - v_i| \le \varepsilon \|v\|_2$. For $\ell_2$ heavy hitters in the turnstile model, queries should be answered with a list $L$ of length $O(1/\varepsilon^2)$ containing all $i \in [d]$ such that $|v_i| \ge \varepsilon \|v\|_2$.*

**Theorem 2.4** (Turnstile $\ell_2$ Point Query [3], Theorem 2 of [23], Lemma 1 of [24]). *There is an algorithm for turnstile $\ell_2$ point query with space complexity $O(\varepsilon^{-2} \log(1/\delta))$, update time $O(\log(1/\delta))$ and query time $O(\log(1/\delta))$, and success probability $1 - \delta$.*

**Theorem 2.5** (Turnstile $\ell_2$ Heavy Hitters [25]). *There is an algorithm for turnstile $\ell_2$ heavy hitters with $O(\varepsilon^{-2} \log n)$ space complexity, $O(\log n)$ update time and $O(\varepsilon^{-2} poly(\log n))$ query time, and success probability $1 - 1/poly(n)$.*

The key new idea of Algorithm 1 is that we implicitly maintain a vector $\widehat{w_t} \in \mathbb{R}^d$ which is updated according to $\widehat{w_{t+1}} \leftarrow (1 - \lambda \eta_t)\widehat{w_t} - \eta_t y_t \ell'(y_t z_t^T R x_t) x_t$. Note that $\widehat{w_t}$ is an approximation to $w_t \in \mathbb{R}^d$ which is obtained by the standard update $w_{t+1} \leftarrow (1 - \lambda \eta_t) - \eta_t y_t \ell'(y_t w_t^T x_t) x_t$. We cannot maintain $\widehat{w_t}$ explicitly, but we give it as input to the linear sketch which we refer to as $\mathcal{A}$ in Algorithm 1. [3] For $\ell_2$ point query, $\mathcal{A}$ is a Countsketch matrix, and for $\ell_2$ heavy hitters it is Expander Sketch [25]. In our QUERY procedure in Algorithm 1, we apply the query procedure of $\mathcal{A}$ to $\mathcal{A}\widehat{w_T}$, where $\widehat{w_T} = \frac{1}{T} \sum_{t=1}^{T} \widehat{w_t}$. This is justified since $\widehat{w_T}$ is a good approximation to $w_*$:

**Theorem 2.6** (Approximating $w_*$ with $\overline{\widehat{w_T}}$). *Let $\varepsilon, \delta \in (0, 1)$. Suppose all of the assumptions in Definition 1.1 hold. Suppose $R$ is a sparse JL matrix with $O(\lambda^{-2} \varepsilon^{-2} \beta^2 \log(dT/\delta) \max(1, \beta/\lambda))$ rows. If $\widehat{w_t}$ is updated according to $\widehat{w_{t+1}} \leftarrow (1 - \lambda \eta_t)\widehat{w_t} - \eta_t y_t \ell'(y_t z_t^T R x_t) x_t$, and $\overline{\widehat{w_t}} = \frac{1}{T} \sum_{t=1}^{T} \widehat{w_t}$, then $\|\overline{\widehat{w_T}} - w_*\|_2 \le \varepsilon \|w_*\|_2$ as long as $T \ge \Omega(\max((\beta^4 H^4)/(\lambda^8 \varepsilon^4 \tau^4), H^4/(\lambda^4 \varepsilon^4 \tau^4)))$.*

**Theorem 2.7** ($\ell_2$ Point Query and Heavy Hitters for Linear Classification). *Suppose $\varepsilon, \delta \in (0, 1)$, all of the assumptions in Definition 1.1 hold, and $T \ge \Omega(\max((\beta^4 H^4)/(\lambda^8 \varepsilon^4 \tau^4), H^4/(\lambda^4 \varepsilon^4 \tau^4)))$.*

---

[3]Here, $\mathcal{A}$ is a linear sketch meaning that, if $\mathcal{A}v$ is itself considered a vector, then $\mathcal{A}v$ is a linear map in terms of $v$. Thus, the update $\widehat{w_{t+1}} \leftarrow (1 - \lambda \eta_t)\widehat{w_t} - \eta_t y_t \ell'(y_t z_t^T R x_t) x_t$ can be implicitly done in sublinear space.

**Algorithm 1** In this algorithm, we give our black-box reduction to $\ell_2$ point query or heavy hitters. Here, $\mathcal{A}$ denotes a linear sketching data structure for $\ell_2$ point query or $\ell_2$ heavy hitters. $\mathcal{A}_t$ denotes the contents of the sketch at time step $t$, and $\overline{\mathcal{A}}$ denotes the sketch containing the average of the contents of $\mathcal{A}_1, \ldots, \mathcal{A}_T$. Since $\mathcal{A}$ is a linear sketch, $\overline{\mathcal{A}}$ can be maintained using only a constant factor more space than that needed to store $\mathcal{A}_t$. Here, QUERY denotes the query procedure of $\mathcal{A}$, that is, the query procedure described in Theorem 2.4 for $\ell_2$ point query and Theorem 2.5 for $\ell_2$ heavy hitters. Note that we can skip the step used in [1] where $z_{t+1}$ is projected onto an $\ell_2$ unit ball, since even without projection, $\|z_{t+1}\|_2 \leq O(H/\lambda)$ by the triangle inequality and induction.

---

**function** INITIALIZATION()
    $R \in \mathbb{R}^{k \times d}$ is a sparse JL matrix with $k = O(\varepsilon^{-2} \log(dT/\delta) \cdot \max(1, \beta/\lambda))$ rows.
    $z_1 \in \mathbb{R}^k$ is set to $0 \in \mathbb{R}^k$.
    The contents of the sketch $\mathcal{A}$ are set to $0 \in \mathbb{R}^d$.
**end function**

**function** UPDATE($x_t, y_t$)
    $z_{t+1} \leftarrow (1 - \lambda \eta_t) z_t - \eta_t y_t \ell'(y_t z_t^T R x_t) R x_t$
    Rescale the contents of $\mathcal{A}$ by $(1 - \lambda \eta_t)$.
    For each nonzero coordinate $i \in [d]$ of $x_t$, update $\mathcal{A}$ according to

$$(i, -\eta_t y_t \ell'(y_t z_t^T R x_t) x_{t,i})$$

**end function**

**function** QUERY()
    $\overline{\mathcal{A}} \leftarrow \frac{1}{T} \sum_{t=1}^T \mathcal{A}_t$
    **Return** QUERY($\overline{\mathcal{A}}$)
**end function**

---

*For $\ell_2$ point query, Algorithm 1 has $O(\lambda^{-2} \varepsilon^{-2} \beta^2 \log(dT/\delta) \max(1, \beta/\lambda) + \varepsilon^{-2} \log(1/\delta))$ space complexity, $O(\lambda^{-1} \varepsilon^{-1} \beta \log(dT/\delta) \max(1, \sqrt{\beta/\lambda}) + \log(1/\delta)) \cdot nnz(x_t)$ update time, $O(\log(1/\delta))$ query time, and success probability $1 - \delta$. For $\ell_2$ heavy hitters, Algorithm 1 has $O(\lambda^{-2} \varepsilon^{-2} \beta^2 \log(dT/\delta) \max(1, \beta/\lambda) + \varepsilon^{-2} \log d)$ space complexity, $O(\lambda^{-1} \varepsilon^{-1} \beta \log(dT/\delta) \max(1, \sqrt{\beta/\lambda}) + \log d) \cdot nnz(x_t)$ update time, $O(\varepsilon^{-2} poly(\log d))$ query time, and success probability $1 - 1/poly(d) - \delta$.*

The proofs of Theorems 2.6 and 2.7 are given in the supplementary. Note that the query times are simply those of CountSketch [3] / ExpanderSketch [25] respectively.

## 3 Deterministic $\ell_1$ Point Query and a Second Algorithm for $\ell_2$ Point Query

We now give a simple deterministic algorithm for $\ell_1$ point query with sublinear space. The algorithm is based on that of [1]. However, the sketching matrix $R$ is now an $\varepsilon$-incoherent matrix:

**Theorem 3.1** ($\varepsilon$-Incoherent Matrices [6]). *Let $n \in \mathbb{N}$ and $\varepsilon > 0$. Then, there exists a matrix $A \in \mathbb{R}^{m \times n}$, where $m = O(\varepsilon^{-2} \min(\log n, (\log n/(\log \log n + \log 1/\varepsilon))^2))$, such that for all $i \in [n]$, $\|A_i\|_2 = 1$, and for all distinct $i, j \in [n]$, $\langle A_i, A_j \rangle \leq \varepsilon$. Moreover, $A$ can be constructed deterministically in $poly(n)$ time. $A$ does not need to be stored explicitly, and each column can be generated on demand in low space (we describe this in the supplementary).*

We first analyze the algorithm in the batch setting, showing that $\|R^T z_* - w_*\|_\infty \leq \varepsilon \|w_*\|_1$, and then using online gradient descent regret bounds to show that the same is true for $\overline{z}$. Our analysis of this algorithm is similar to [1] — however, using the recovery procedure in Algorithm 2 leads to an improved space complexity compared to [1] (here we assume $\|x_t\|_1 \leq \gamma$ as in [1]):

**Theorem 3.2** (Analysis of $\ell_1$ Point Query on $w_*$ with Incoherent Matrix). *Suppose all of the assumptions in Definition 1.1 hold, $\|x_t\|_1 \leq \gamma$ for all $t \in [T]$, and there exists some constant $\theta > 0$ independent of $T$ such that $\|w_*\|_1 \geq \theta$. If $R$ and $\overline{z}$ are defined as in Algorithm 2, with $R$ being an incoherent matrix, then $\|R^T \overline{z} - w_*\|_\infty \leq \varepsilon \|w_*\|_1$, as long as $T \geq \Omega(H^4(1 + \sqrt{\varepsilon}\gamma)^4/(\lambda^4 \varepsilon^4 \theta^4))$.*

**Algorithm 2** Algorithm for $\ell_1$ point query and $\ell_2$ point query. For $\ell_1$ point query, $R$ is an incoherent matrix with $O(\varepsilon^{-2} \log d \cdot \max(1, \gamma^2 \beta/\lambda))$ rows, while for $\ell_2$ point query, $R$ is a sparse JL matrix with $O(\varepsilon^{-2} \log(dT/\delta) \max(1, \beta/\lambda))$ rows.

---

1: **function** INITIALIZATION()
2:     $R \in \mathbb{R}^{k \times d}$ is defined as in the caption.
3:     $z_1 \in \mathbb{R}^k$ is initially set to 0.
4: **end function**

5: **function** UPDATE($x_t, y_t$)
6:     $z_{t+1} \leftarrow (1 - \lambda \eta_t) z_t - \eta_t y_t \ell'(y_t z_t^T R x_t) R x_t$
7: **end function**

8: **function** ESTIMATE-WEIGHTS($i$)
9:     $\overline{z_T} \leftarrow \frac{1}{T} \sum_{t=1}^{T} z_t$
10:    **return** $R_i^T \overline{z_T}$
11: **end function**

---

If $R$ is instead a sparse JL matrix with $O(\varepsilon^{-2} \log(dT/\delta) \max(1, \beta/\lambda))$ rows, Algorithm 2 gives an $\ell_2$ point query algorithm. Note that for $\ell_2$ point query, the space complexity of Algorithm 2 has a better dependence on $\lambda$ compared to Algorithm 1.

**Theorem 3.3** ($\ell_2$ Point Query using only a JL Matrix). *Let $\varepsilon, \delta \in (0, 1)$, and suppose all of the assumptions in Definition 1.1 hold. If $R$ and $\overline{z}$ are defined as in Algorithm 2, with $R$ being a sparse JL matrix, then $\|R^T \overline{z} - w_*\|_\infty \leq \varepsilon \|w_*\|_2$ with probability $1 - \delta$, as long as $T \geq \Omega(H^4/(\lambda^4 \varepsilon^4 \tau^4))$.*

## 4 Low Rank Tensor Classification and Kernel Classification

We next consider $\ell_2$ point query and heavy hitters in the case where $x_t \in \mathbb{R}^{d^p}$ is a $p^{th}$ order tensor, of rank at most $k$, and is provided as the sum of $k$ rank-1 tensors. This is motivated by polynomial kernel classification as well as other applications in classification with tensor inputs mentioned above. Our main tool will be a JL matrix which can be quickly applied to outer products of vectors [11]:

**Theorem 4.1** (Recursive Tensor Sketch — Follows from the Proof of Theorem 2 of [11]). *Let $n, p, d \in \mathbb{N}$, $\varepsilon, \delta > 0$. Then, there is a random matrix $R \in \mathbb{R}^{m \times d^p}$, with $m = \Theta(\varepsilon^{-2} p \log(1/\varepsilon\delta)^3)$, such that for $x, y \in \mathbb{R}^{d^p}$, $\Pr_M[|\langle Rx, Ry \rangle - \langle x, y \rangle| \geq \varepsilon \|x\|_2 \|y\|_2] \leq 1 - \delta$ and for $x_1, x_2, \ldots, x_p \in \mathbb{R}^d$, $R(x_1 \otimes x_2 \otimes \ldots \otimes x_p)$ can be computed in $poly(\varepsilon^{-1} p \log(1/\delta)) \sum_{i=1}^p nnz(x_i)$ time.* [4]

This immediately gives us an algorithm for $\ell_2$ point query, since in Algorithm 2, $R$ can be replaced by any JL matrix. Note that the query procedure can be done in $poly(p \log(dT/\delta)/\varepsilon)$ time (up to problem-dependent parameters), since for $i = (i_1, \ldots, i_p) \in [d]^p$, $R_i = R(e_{i_1} \otimes \ldots \otimes e_{i_p})$ can be computed in $poly(p \log(dT/\delta)/\varepsilon)$ time. For completeness, we explicitly give the pseudocode for this algorithm in the supplementary — the guarantees of this algorithm are stated below:

**Theorem 4.2** (Tensor Classification Point Query). *Let $\varepsilon, \delta \in (0, 1)$ and suppose all of the assumptions in Definition 1.1 hold. Let $R$ be the JL matrix of Theorem 4.1, with $poly(\varepsilon^{-1} p \log(dT/\delta)(1 + \beta/\lambda))$ rows, and suppose $T \geq \Omega(H^4/(\lambda^4 \varepsilon^4 \tau^4))$. Then, with probability $1 - \delta$, $\|R^T \overline{z} - w_*\|_\infty \leq \varepsilon \|w_*\|_2$. Thus, there is an algorithm for $\ell_2$ point query on $w_*$ with space complexity and query time $poly(\varepsilon^{-1} p \log(dT/\delta)(1 + \beta/\lambda))$, and update time $poly(\varepsilon^{-1} p \log(dT/\delta)(1 + \beta/\lambda)) \sum_{i=1}^k \sum_{j=1}^p nnz(x_t^{(i,j)}) \leq kd \, poly(\varepsilon^{-1} p \log(dT/\delta)(1 + \beta/\lambda))$.*

Next we consider $\ell_2$ heavy hitters on $w_* \in \mathbb{R}^{d^p}$. To simplify the problem, we reduce to the setting where we are given $v \in \mathbb{R}^{d^p}$ and $v$ is given updates of the form $v \leftarrow v + x_1 \otimes \ldots \otimes x_p$ (where we are given $x_1, \ldots, x_p$). This reduction is valid by Theorem 2.6, since updates to $\widehat{w_t}$ are of this form. Our algorithm for this setting is shown in Algorithm 3, with the following guarantees:

---

[4] $R(x_1 \otimes x_2 \otimes \ldots \otimes x_p)$ can be computed in one pass over the nonzero entries of $x_1, x_2, \ldots, x_p$: by the construction of [11], $R$ is essentially a tree of sketching matrices, with $2p - 1$ nodes. At the base of this tree are matrices $R^1, \ldots, R^p$, which can be separately applied to $x_1, \ldots, x_p$ respectively — from this point, only $poly(\varepsilon^{-1} p \log(d/\delta))$ space is needed to finish the computation.

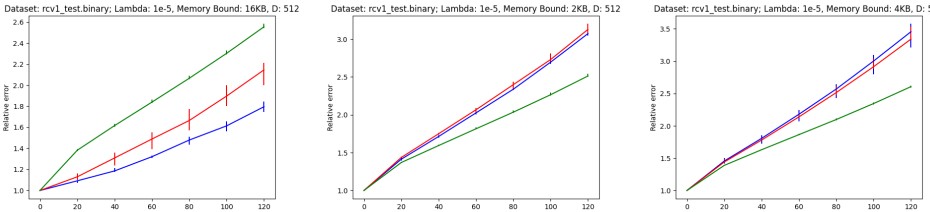

Figure 1: Blue is Algorithm 2, green is Algorithm 1, and red is the algorithm of [1]. We show the median of 5 trials, with error bars showing the smallest and largest relative errors across those trials.

**Theorem 4.3** (Tensor $\ell_2$ Heavy Hitters Algorithm). *Let $\varepsilon, \delta \in (0,1)$. Let $v \in \mathbb{R}^{d^p}$, which can be incrementally updated by a rank-1 tensor. Then, Algorithm 3 returns a list containing all $(i_1, i_2, \ldots, i_p) \in [d]^p$ such that $|v(i_1, \ldots, i_p)| \geq \varepsilon \|v\|_2$. The space complexity is $poly(\varepsilon^{-1} p \log(d/\delta))$, the query time is $poly(\varepsilon^{-1} p \log(d/\delta))$, and the time needed for the update $v \leftarrow v + x$, for a rank-1 tensor $x = x_1 \otimes \ldots \otimes x_p$, is $poly(\varepsilon^{-1} p \log(d/\delta)) \sum_{j=1}^{p} nnz(x_j)$.*

Theorem 4.3 implies that, by adapting Algorithm 1, but with $R$ being the JL matrix of Theorem 4.1 instead of a sparse JL matrix, we can get an algorithm for heavy hitters in the linear classification setting, where the $x_t$ are rank-$k$ tensors, and with space and time complexity polynomial in $p$:

**Theorem 4.4** (Tensor Classification $\ell_2$ Heavy Hitters). *Let $\varepsilon, \delta \in (0,1)$. Suppose all assumptions in Definition 1.1 hold, and $T \geq \Omega(\max((\beta^4 H^4)/(\lambda^8 \varepsilon^4 \tau^4), H^4/(\lambda^4 \varepsilon^4 \tau^4)))$. Then, there is an algorithm for $\ell_2$ heavy hitters on $w_*$ with space complexity $poly(\varepsilon^{-1} p \log(dT/\delta)(1 + \beta/\lambda))$, query time $poly(\varepsilon^{-1} p \log(d/\delta))$, and update time $poly(\varepsilon^{-1} p \log(dT/\delta)(1 + \beta/\lambda)) \sum_{i=1}^{k} \sum_{j=1}^{p} nnz(x_t^{(i,j)}) \leq kd \cdot poly(\varepsilon^{-1} p \log(dT/\delta)(1 + \beta/\lambda))$.*

As corollaries, we give results for heavy hitters for kernel classification in the supplementary.

## 5  Point Query Experiments

We compare Algorithms 1 and 2 (where $R$ is a JL matrix for Algorithm 2) with the algorithm of [1].

**Datasets:** We use the following datasets (which were also used in [1]): RCV1 [13], KDD Cup 2010 Algebra [27] (we use a transformed version due to [28]), and the malicious URL dataset of [29].

**Parameters:** We perform online logistic regression $\lambda \in \{10^{-3}, 10^{-4}, 10^{-5}\}$. Each algorithm is subject to a memory constraint $M \in \{2\,KB, 4\,KB, 8\,KB, 16\,KB, 32\,KB\}$. For Algorithm 2 and the algorithm of [1], the dimensions of the JL/Countsketch matrices for a particular memory constraint are given by the corresponding row in Table 1 of [1] (noted in [1] to be the best performing configurations for Countsketch). For Algorithm 1, which uses a JL matrix and a separate Countsketch matrix, we also use the same configuration for both matrices, but with the width divided by 2.

**Error Metric:** We compare the algorithms in terms of how well they recover the top weights of $w_*$. We use a similar relative error metric to that used in Subsection 7.2 of [1]. To estimate how well one of the three algorithms recovers the top $K$ weights, we let $w^K$ be the $K$-sparse vector whose entries are the $K$ largest estimated coordinates obtained by this algorithm. We let $w_*^m$ be the $m$-sparse vector whose entries are the $m$ largest coordinates of $w_*$, for any $m$. Then, we use $\|w^K - w_*^D\|_2 / \|w_*^K - w_*^D\|_2$ as our metric, where $D = 512 \gg K$. This is similar to the metric used in [1] (that is, $\|w^K - w_*\|_2 / \|w_*^K - w_*\|_2$) — we use $w_*^D$ instead since this omits the smaller weights and might therefore better measure how well the algorithms recover the larger weights. We also note that in place of $w_*$, we use the weight vector that is obtained by online logistic regression for these experiments — this was also done by [1] in their experiments.

**Results:** Here we show a few plots on the RCV1 dataset in Figure 5, with $\lambda = 10^{-5}$ — all plots are in the supplementary. With a memory budget of 16 KB, when $K = 120$, Algorithm 2 gives an improvement of roughly 15% over the algorithm of [1], which in turn performs better than Algorithm 1. With memory budgets of 2 KB and 4 KB, Algorithm 1 has the best performance.

**Algorithm 3** Algorithm for $\ell_2$ heavy hitters (i.e., without classification) where the input is $v \in \mathbb{R}^{d^p}$ which is updated according to $v \leftarrow v + x$, where $x = x_1 \otimes \ldots \otimes x_p$. For ease of presentation, we do not distinguish between a sketch $S : \mathbb{R}^a \to \mathbb{R}^b$ (i.e., $S \in \mathbb{R}^{b \times a}$) and its contents $Sv \in \mathbb{R}^b$ (for $v \in \mathbb{R}^a$). We make use of a standard $\ell_2$ heavy hitters data structure $\text{ONEMODESKETCH}^{(i)}$, whose size has a logarithmic dependence on $1/\delta$ — while such a dependence is not stated by [25] (which achieves the optimal space complexity for $\ell_2$ heavy hitters), we use the dyadic trick, which still has sublinear time and space complexity — see Theorem 1 of [26].

**Require:** $\varepsilon, \delta \in (0, 1)$
**Ensure:** Return a list $L \subset [d]^p$ with $|L| \leq O(1/\varepsilon^2)$ containing all $i \in [d]^p$ such that $|v_i| \geq \varepsilon \|v\|_2$.

**function** INITIALIZATION()
    — For each $i \in [p]$, $\text{COMPRESSOTHERMODES}^{(i)} : \mathbb{R}^{d^{p-1}} \to \mathbb{R}^{\text{poly}(p \log(d/\delta))}$ is the sketch of [11], with $\varepsilon = O(1)$.
    — For each $i \in [p]$, $\text{ONEMODESKETCH}^{(i)} : \mathbb{R}^{d \cdot \text{poly}(p \log(d/\delta))} \to \mathbb{R}^{\text{poly}(\log(d/\delta)/\varepsilon)}$ is a usual $\ell_2$ heavy hitter data structure (such as that of [25]) with accuracy $\varepsilon' = \frac{\varepsilon}{\text{poly}(p \log(d/\delta))}$.
    — For each $i \in [p-1]$, $\text{COMPRESSSUFFIX}^{(i)} : \mathbb{R}^{d^{p-i}} \to \mathbb{R}^{\text{poly}(p \log(d/\delta))}$ is the sketch of [11] with $p - i$ in place of $p$ and $\varepsilon = O(1)$.
    — For each $i \in [p-1]$, $\text{PREFIXPOINTQUERY}^{(i)} : \mathbb{R}^{d^i \cdot \text{poly}(p \log(d/\delta))} \to \mathbb{R}^{\text{poly}(p \log(d/\delta)/\varepsilon)}$ is the sketch of [11] with the first $i$ input modes being $d$-dimensional and the last mode being $\text{poly}(p \log(d/\delta))$ dimensional, with accuracy $\varepsilon' = \frac{\varepsilon}{\text{poly}(p \log(d/\delta))}$. $\text{PREFIXPOINTQUERY}^{(p)}$ from $\mathbb{R}^{d^p}$ to $\mathbb{R}^{\text{poly}(p \log(d/\delta)/\varepsilon)}$ is simply the sketch of [11].
**end function**

// Here we allow $x$ to be a rank-1 tensor without loss of generality. The case where $x$ is a rank-$k$
// tensor is the same, except the update time increases by a factor of $k$.
**function** UPDATE($x = x_1 \otimes \ldots \otimes x_p$)
    — For each $i \in [p]$, update $\text{ONEMODESKETCH}^{(i)}$ by

$$x_i \otimes \text{COMPRESSOTHERMODES}^{(i)}(x_1 \otimes \ldots \otimes x_{i-1} \otimes x_{i+1} \otimes \ldots \otimes x_p)$$

    — Update $\text{PREFIXPOINTQUERY}^{(p)}$ by $x_t$. For $i \in [p-1]$ update $\text{PREFIXPOINTQUERY}^{(i)}$ by

$$x_1 \otimes \ldots \otimes x_i \otimes \text{COMPRESSSUFFIX}(x_{i+1} \otimes \ldots \otimes x_p)$$

**function** QUERY()
    — For each $i \in [p]$, find all $\frac{\varepsilon}{\text{poly}(p \log(d/\delta))}$-heavy hitters $(j, k) \in [d] \times [\text{poly}(p \log(d/\delta)]$ from $\text{ONEMODESKETCH}^{(i)}$.
    — Collect a list $L_i$ of length at most $\text{poly}(\varepsilon^{-1} p \log(d/\delta))$, of coordinates $j \in [d]$ such that $(j, k)$ was returned by $\text{ONEMODESKETCH}^{(i)}$ in the previous step, for some $k \in [\text{poly}(p \log(d/\delta))]$. Note that $L_i$ contains all coordinates in the $i^{th}$ mode potentially comprising an $\varepsilon$ fraction of $\|w_*\|_2$.
    — $L \leftarrow L_1$ // initial list of prefixes of heavy hitters
    — $L \leftarrow$ top $O(1/\varepsilon^2)$ elements of $L$ according to JL-based point query on $\text{PREFIXPOINTQUERY}^{(1)}$.
    **for** $i = 2, \ldots, p$ **do**
        — $L' \leftarrow L \times L_i = \{((j_1, j_2, \ldots, j_{i-1}), j) \mid (j_1, \ldots, j_{i-1}) \in L, j \in L_i\}$.
        — $L \leftarrow$ top $O(1/\varepsilon^2)$ elements of $L'$ according to JL-based point query on $\text{PREFIXPOINTQUERY}^{(i)}$.
    **end for**
    **Return** $L$
**end function**

## Acknowledgments and Disclosure of Funding

D. Woodruff was supported by NSF CCF-1815840, Office of Naval Research grant N00014-18-1-2562, and a Simons Investigator Award.

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
