# OpenReview forum: "Linear and Kernel Classification in the Streaming Model: Improved Bounds for Heavy Hitters"
_NeurIPS.cc/2021/Conference — NeurIPS 2021 Poster_

### Official Review · Reviewer_LYkE · 2021-07-05

**Rating:** 6
**Confidence:** 4

**Summary:**

This paper tackles a very basic problem in machine learning, linear (and polynomial, and kernel) classification in the streaming model.  Such learned models can be represented as a coefficients vector w.  The task studied in this paper is to approximately identify the largest coefficients in the optimal coefficient vector w.

This model was introduced in a SIGMOD 2018 paper.  This paper:
 - improves bounds under the l_1 model, including a deterministic option
 - provides new poly(dimension) bounds in the l_2 model
 - removes assumption that data is random ordered, it allows adversarial order
 - extends to polynomial and kernel SVM classifiers

**Limitations And Societal Impact:**

As this is a theory paper, the authors do not really engage with this aspect.  It mostly removes limitations from prior work.

As mentioned I am concerned with the focus on explainability by high weights -- but more a modeling issue than a serious societal one rooted in this paper.

**Main Review:**

If one accepts the model as important, these are sizable improvements.  They are achieved by layering many recent techniques on this setting.  The ideas do not appear entirely new (from a techniques point of view), but the application of them is nonetheless non-trivial and varied.

However, I am not enthusiastic about the model.  It perpetuates the fallacy that high-weight coefficients are the most important elements of a model.  This may not be true if variables are not properly normalized (if that is even possible).  For instance, the even more involved Shapley values are not necessarily as powerful as often claimed (https://arxiv.org/abs/2002.11097).
For instance, the paper does not make any claims about how well the compressed model approximates the loss function or generalizes to new data under any assumptions.

Hence, while I like the technical improvements provided in this work, I am skeptical of the overall premise.

**Time Spent Reviewing:**

3

---

> ### Author Response · Authors · 2021-08-10
> **Response to Reviewer LYkE**
>
> Thank you very much for your review. We respond to the concerns raised below.
>
> $\textbf{Theoretical Guarantees for Classification Error:}$ The theoretical guarantees for classification error, when using the compressed vector, can be obtained as follows, using the JL lemma. Note that at iteration $T$, if we use the average iterate for the compressed weight vector $\overline{z} = \frac{1}{T} \sum_{t = 1}^T z_t$ for prediction, then $\overline{z}$ is approximately equal to the optimal compressed weight vector $z_*$ for the batch setting, for large $T$. In addition, as shown in the supplementary material,
>
> $ \|z_* - Rw_*\|_2 $
>
>  <=  $(2\beta /\lambda)^{1/2} F \|w_*\|_2$
>
> <= $O(\epsilon \|w_*\|_2) $
>
> where in the last inequality we omit problem-dependent parameters here. Finally, applying the JL lemma (see Lemma A.3 in the supplementary material) we find that
>
> $\langle Rw_*, Rx_t \rangle = \langle w_*, x_t \rangle \pm O(\epsilon \|w_*\|_2)$
>
> Thus, assuming the loss functions $f_t$ are $H$-Lipschitz for some constant $H$, we find that
>
> $\frac{1}{T} \sum_{t = 1}^T f(y_t \overline{z}^T Rx_t) = \frac{1}{T} \sum_{t = 1}^T f(y_t w_*^T x_t) \pm O(\epsilon \|w_*\|_2)$
>
> Also note that the work "Sketching Linear Classifiers over Data Streams'' by Tai, et al. gives experimental results (see Figure 6 of that work) showing that their compressed classifier achieves a similar classification error rate to that of unsketched online logistic regression.
>
> $\textbf{Space Lower Bounds for Heavy Hitters:}$ By Theorem 4.3 of "A Simple Proof of a New Set Disjointness with Applications to Data Streams'' by Kamath, et al., at least the minimum of $\frac{\log(1/\delta)}{\epsilon^2}$  and $d$ space is needed even for standard $\ell_2$ heavy hitters (that is, for finding the coordinates $i$ such that $|w_i| \geq \epsilon |w|_2$).
>
> In particular, to obtain estimates of all the features in a stream, at least poly$(d)$ space would be needed. It is thus reasonable to expect that, without additional assumptions, finding the $\epsilon$-heavy hitters of $w_*$ is the best one could do in a streaming setting, if the goal is to obtain sublinear space in $d$.
>
> $\textbf{Technical Novelties:}$ Here we highlight the additional techniques we introduce in this work, which go significantly beyond direct applications of known techniques in the streaming literature, even beyond the applications to classification/regression:
>
> - In Section 2 we give a black box reduction from $\ell_2$ point query/heavy hitters in the linear classification setting, to standard $\ell_2$ point query/heavy hitters. The key technique introduced in this section is the black box reduction using JL matrices, which itself was not previously known (e.g. see the previous works on related problems, such as "Sketching Linear Classifiers over Data Streams'' by Tai, et al. and "MISSION: Ultra Large-Scale Feature Selection using Count-Sketches'' by Aghazadeh, et al.).
>
> - In Section 4, we show that just a single JL matrix suffices to find the $\ell_2$ heavy hitters of $w_*$. This was not previously known even in the streaming literature --- that is, even for standard $\ell_2$ heavy hitters.
>
> - In Section 5, we introduce an algorithm for polynomial kernel logistic regression, and logistic regression where the inputs are low-rank tensors. To find the heavy hitters in this setting, it is not possible to directly apply standard streaming techniques without incurring a $d^q$ time complexity --- thus we introduce the result given in Theorem 5.2 (of which the main component is Algorithm 4 in the supplementary material, which we could not include in the main paper due to space constraints). Though this algorithm makes use of the result from the work "Oblivious Sketching of High-Degree Polynomial Kernels'' by Ahle, et al., Algorithm 4 is not a direct application of that work, and introduces several new ideas. The key new idea in Algorithm 4 is that of iteratively forming prefixes of the $\ell_2$ heavy hitters in $[d]^q$, mode by mode.

---

> > ### Comment · Reviewer_LYkE · 2021-08-23
> > **lipschitz loss functions**
> >
> > OK.  I like the argument based on the potential H-Lipschitz properties of the loss function.  I think a bound based on this statement (I'd like to see how H factors into the final bound in a stated theorem in a final version) helps clarify the usefulness of this work, and its limitations.
> >
> > As stated, I am also a fan of the technical contributions of the paper.
> >
> > I will raise my score from 5 to 6.

---

> > > ### Author Response · Authors · 2021-08-30
> > > **Test Accuracy Using Only Top Weights**
> > >
> > > Thank you very much for your comment! We will include that argument in the final version if accepted.
> > >
> > > Regarding your other concern about our model perpetuating the fallacy that high-weight coefficients are the most important elements of a model, we found experimentally that on the RCV1 dataset used in our paper, the high weight coefficients are in fact enough to obtain a good test error. We divided the RCV1 dataset equally into two parts - the first part was used as the training dataset and the second was used as the test dataset. For training, we obtained the weight vector using online logistic regression, and to obtain the test error, we zeroed out all but the top K coordinates of the weight vector (in terms of the absolute value) for various values of K, and used this K-sparse weight vector for prediction. We obtained the following results:
> > >
> > > Test accuracy using top  50  weights:  0.8555949217596693
> > >
> > > Test accuracy using top  60  weights:  0.8679362267493357
> > >
> > > Test accuracy using top  70  weights:  0.8767581930912312
> > >
> > > Test accuracy using top  80  weights:  0.8782285208148805
> > >
> > > Test accuracy using top  90  weights:  0.8811662237968704
> > >
> > > Test accuracy using top  100  weights:  0.8952583407144966
> > >
> > > Test accuracy using top  200  weights:  0.9232122822556835
> > >
> > > Test accuracy using top  300  weights:  0.9370711544139356
> > >
> > > Test accuracy using top  400  weights:  0.9396220844405079
> > >
> > > Test accuracy using top  500  weights:  0.942849129022734
> > >
> > > Test accuracy using top  600  weights:  0.9442810746973723
> > >
> > > Test accuracy using top  700  weights:  0.9470386772955418
> > >
> > > Test accuracy using top  800  weights:  0.9481842338352524
> > >
> > > Test accuracy using top  900  weights:  0.9491378801299085
> > >
> > > Test accuracy using top  1000  weights:  0.9501505757307351
> > >
> > > Test accuracy using top  10000  weights:  0.957472689695896
> > >
> > > Test accuracy using top  20000  weights:  0.9573516386182462
> > >
> > > Test accuracy using top  40000  weights:  0.9573693534100974
> > >
> > > Note that the number of nonzero weights in the true model (without sparsifying) is 41130, and the test accuracy using the full model is 0.9573693534100974.
> > >
> > > We would also like to reiterate that in general, in a stream, finding the heavy hitters is the best that can be done in sublinear space, as we discussed in our previous comment to you.

---

> > > > ### Comment · Reviewer_LYkE · 2021-08-30
> > > > **top weights**
> > > >
> > > > Yes, I understand that on some, perhaps even many, data sets that the top weights are the most useful.  But it is not *always* the case.  And that implication can lead to serious harms due to overconfidence in being able to interpret models when it is made generally.

---

> > > > > ### Author Response · Authors · 2021-08-30
> > > > > **top weights discussion in paper**
> > > > >
> > > > > Thanks for the comment! We will add discussion about this not always being the case to the introduction of our paper.

---

### Official Review · Reviewer_6Mw1 · 2021-07-08

**Rating:** 5
**Confidence:** 4

**Summary:**

The focus of this paper is one linear and kernel classification in a streaming setting with limited working memory. I will focus on linear classification in this summary. Given a loss function l and a set of training samples {(x_t,y_t)}_{t=1}^T, the goal is to find the vector w minimizing $(1/T) \sum_{t=1}^T l(y_t w^T x_t) + \lambda\|w\|_2^2$. The twist is that this must be done using sublinear memory. Here sublinear is also in d - the number of features. Thus the optimal vector w cannot even be stored in memory.

What one does instead is to use techniques from streaming algorithms to extract the large coordinates of w, hoping that most of w is captured in a few coordinates of large magnitude. Formally, the setup is that training data arrives in a streaming fashion, with each item in the stream representing a coordinate of an x_t along with y_t. So we only see one single coordinate at a time.

In a standard streaming setting, one can use classic sketches such as the CountSketch to compute a small-memory representation of a given vector w, while allowing finding the heavy hitters (coordinates of magnitude at least $\varepsilon \|w\|_2$) efficiently. The tricky part in the problem considered in this paper, is to simultanously find the small coordinates AND figure out what the optimal solution w is.

A standard solution to linear classification with various loss functions, is to use online gradient descent. This however requires maintaining the current vector of parameters w. This is too expensive here, and the main idea in the work is to sketch w using one of the classic sketches for turnstile streaming heavy hitters / point queries. One can then "simulate" gradient descent via a gradient descent on the parameters in the sketch.

The results are also extended to tensor and kernel regression.

**Limitations And Societal Impact:**

Yes

**Main Review:**

To summarize my impression of the paper, I find that the most questionable part is the motivation for studying this problem. How often do you really have so many features that you do not have sufficient memory to store them? And more importantly, when you do, how often are the coordinates of the optimal solution w collected on a few coordinates of very large magnitude? In particular this last point is not even discussed in the paper. If there are no eps-heavy hitters in w, then the heavy-hitters guarantees are meaningless. The paper also does not include experiments that show whether this happens on some real-life data sets. Instead the experiments only focus on the ratio between the quality of the solution found by the new algorithm and the solution found by a previous algorithm. Which I agree is an important experiment. But another very interesting value to know would be the ratio $\|w^K - w*\|_2/\|w*\|_2$ on practical data sets. It is not clear at all whether this number is close to 1 or to 0. If it is very close to 1, then the heavy hitters seems a bit meaningless.

The above being said, I think the solution is a clean combination of known techniques from turnstile streaming. It elegantly uses linearity of the turnstile streaming sketches and combines it nicely with gradient descent. The writing of the paper is pretty dense and there are also some places where notation is not introduced (like in the main algorithm, there is an A_t which is never defined, and one has to guess that QUERY(A_T) inside QUERY is a call to a sketch's query algorithm). There are also a number of polynomial dependencies on H lambda eps tau that are pretty nasty from a practical point of view.

All in all, the paper combines standard techniques from streaming to solve a low-space version of a fundamental problem. I'm not completely convinced of the importance of the problem, and the one experiment shown in the paper is not super-convincing. I thus find that the paper fall a bit short of the bar for NeurIPS.


**Time Spent Reviewing:**

3

---

> ### Author Response · Authors · 2021-08-10
> **Response to Reviewer 6Mw1**
>
> Thank you very much for your review. We respond below to the concerns you raised.
>
> $\textbf{Problems with High-Dimensional Feature Spaces:}$ As mentioned in the second paragraph of the introduction in "MISSION: Ultra Large-Scale Feature Selection using Count-Sketches'' by Aghazadeh, et al., there are several computational biology problems whose space requirements are too much for a single computer.
>
> In addition, one additional motivation for obtaining space requirements that are sublinear in $d$, where $d$ is the number of features, is the setting of small devices with memory constraints (where one simply wants to maintain a model on a small device locally, without any communication with a central server) --- this is the motivation for "Sketching Linear Classifiers over Data Streams'' by Tai, et al. This is discussed further in the second and third paragraphs of the introduction of that work, where it is noted that maintaining n-gram features can lead to space requirements that are too large for a small device.
>
> $\textbf{Sparsity of Optimal Weight Vector in Practice:}$
> We computed the optimal weight vector $w$ using online logistic regression on the Malicious URL dataset, used in our work, and in the paper "Sketching Linear Classifiers over Data Streams'' by Tai, et al. In this dataset, the training examples have 3,230,000 features. We observed that when $\lambda = \frac{1}{1000}$, then $\frac{\|w_k - w\|_2}{\|w\|_2} < 0.2$, where $k = 20,000$. Even when $\lambda = 10^{-4}$, the ratio $\frac{\|w_k - w\|_2}{\|w\|_2}$ is only slightly larger than $0.2$ for $k = 20,000$, and when $\lambda = 10^{-5}$, the ratio $\frac{\|w_k - w\|_2}{\|w\|_2}$ is roughly $0.4$ for $k = 20,000$.
>
> We view the above empirical results as evidence that there are practical data sets where $w$ is well-approximated by a sparse vector.
>
> Thank you also for the comments on the presentation - if accepted, we can use the extra page to make the paper less dense. We will also explain $A_t$ and QUERY$(A_T)$, thank you for pointing that out.
>
> $\textbf{Technical Novelties:}$ Here we highlight the additional techniques we introduce in this work, which go significantly beyond direct applications of known techniques in the streaming literature, even beyond the applications to classification/regression:
>
> - In Section 2 we give a black box reduction from $\ell_2$ point query/heavy hitters in the linear classification setting, to standard $\ell_2$ point query/heavy hitters. The key technique introduced in this section is the black box reduction using JL matrices, which itself was not previously known (e.g. see the previous works on related problems, such as "Sketching Linear Classifiers over Data Streams'' by Tai, et al. and "MISSION: Ultra Large-Scale Feature Selection using Count-Sketches'' by Aghazadeh, et al.).
>
> - In Section 4, we show that just a single JL matrix suffices to find the $\ell_2$ heavy hitters of $w_*$. This was not previously known even in the streaming literature --- that is, even for standard $\ell_2$ heavy hitters.
>
> - In Section 5, we introduce an algorithm for polynomial kernel logistic regression, and logistic regression where the inputs are low-rank tensors. To find the heavy hitters in this setting, it is not possible to directly apply standard streaming techniques without incurring a $d^q$ time complexity --- thus we introduce the result given in Theorem 5.2 (of which the main component is Algorithm 4 in the supplementary material, which we could not include in the main paper due to space constraints). Though this algorithm makes use of the result from the work ``Oblivious Sketching of High-Degree Polynomial Kernels'' by Ahle, et al., Algorithm 4 is not a direct application of that work, and introduces several new ideas. The key new idea in Algorithm 4 is that of iteratively forming prefixes of the $\ell_2$ heavy hitters in $[d]^q$, mode by mode.

---

> > ### Author Response · Authors · 2021-08-30
> > **Test Accuracy Using Only Few Highest Weights**
> >
> > Regarding your concern that the mass of the weight vector might not concentrate on the top K weights for K small, we found experimentally that on the RCV1 dataset used in our paper, the high weight coefficients are in fact enough to obtain a good test error - this is arguably just as important as having most of the mass of the weight vector concentrated on the top K weights.
> >
> > We divided the RCV1 dataset equally into two parts - the first part was used as the training dataset and the second was used as the test dataset. For training, we obtained the weight vector using online logistic regression, and to obtain the test error, we zeroed out all but the top K coordinates of the weight vector (in terms of the absolute value) for various values of K, and used this K-sparse weight vector for prediction. We obtained the following results:
> >
> > Test accuracy using top 50 weights: 0.8555949217596693
> >
> > Test accuracy using top 60 weights: 0.8679362267493357
> >
> > Test accuracy using top 70 weights: 0.8767581930912312
> >
> > Test accuracy using top 80 weights: 0.8782285208148805
> >
> > Test accuracy using top 90 weights: 0.8811662237968704
> >
> > Test accuracy using top 100 weights: 0.8952583407144966
> >
> > Test accuracy using top 200 weights: 0.9232122822556835
> >
> > Test accuracy using top 300 weights: 0.9370711544139356
> >
> > Test accuracy using top 400 weights: 0.9396220844405079
> >
> > Test accuracy using top 500 weights: 0.942849129022734
> >
> > Test accuracy using top 600 weights: 0.9442810746973723
> >
> > Test accuracy using top 700 weights: 0.9470386772955418
> >
> > Test accuracy using top 800 weights: 0.9481842338352524
> >
> > Test accuracy using top 900 weights: 0.9491378801299085
> >
> > Test accuracy using top 1000 weights: 0.9501505757307351
> >
> > Test accuracy using top 10000 weights: 0.957472689695896
> >
> > Test accuracy using top 20000 weights: 0.9573516386182462
> >
> > Test accuracy using top 40000 weights: 0.9573693534100974
> >
> > Note that the number of nonzero weights in the true model (without sparsifying) is 41130, and the test accuracy using the full model is 0.9573693534100974.
> >
> > In the worst case, finding the top weights (i.e. the heavy hitters) is the best that can be done in a stream, in sublinear space:
> >
> > $\textbf{Space Lower Bounds for Heavy Hitters:}$ By Theorem 4.3 of "A Simple Proof of a New Set Disjointness with Applications to Data Streams'' by Kamath, et al., at least the minimum of $\frac{\log(1/\delta)}{\epsilon^2}$  and $d$ space is needed even for standard $\ell_2$ heavy hitters (that is, for finding the coordinates $i$ such that $|w_i| \geq \epsilon |w|_2$).
> >
> > In particular, to obtain estimates of all the features in a stream, at least poly$(d)$ space would be needed. It is thus reasonable to expect that, without additional assumptions, finding the $\epsilon$-heavy hitters of $w_*$ is the best one could do in a streaming setting, if the goal is to obtain sublinear space in $d$.

---

### Official Review · Reviewer_PPCJ · 2021-07-16

**Rating:** 6
**Confidence:** 3

**Summary:**

The submission is a follow up paper on prior works on performing online gradient descent in sketch space.   The premise here is that the dimension d is very large and therefore we do not want to maintain the parameters vector w* explicitly.   We may only want to know the heavy hitter entries of the vector or to be able to query for values at particular indices (with error of the order of a fraction of the norm).  The approach is to use sketches to "update" w*.

The submission improves over prior work using a “plug and play” approach of taking known results and methods from the streaming literature and applying them in the online gradient descent context.

**Ethical Concerns:**

I did not identify ethical concerns.

**Limitations And Societal Impact:**

Yes.

**Main Review:**

Strength: Natural problem that  was already highlighted in prior work and the submission improves over the results.

Weaknesses:
The paper presentation is not easily accessible to people not familiar with the particular prior streaming literature.

The solution is essentially a simply plug and play taking prior tools from streaming, adopted to the online gradient descent framework and the parameters vectors w*.


**Time Spent Reviewing:**

1

---

> ### Author Response · Authors · 2021-08-10
> **Response to Reviewer PPCJ**
>
> $\textbf{Technical Novelties:}$ Here we highlight the additional techniques we introduce in this work, which go significantly beyond direct applications of known techniques in the streaming literature, even beyond the applications to classification/regression:
>
> - In Section 2 we give a black box reduction from $\ell_2$ point query/heavy hitters in the linear classification setting, to standard $\ell_2$ point query/heavy hitters. The key technique introduced in this section is the black box reduction using JL matrices, which itself was not previously known (e.g. see the previous works on related problems, such as "Sketching Linear Classifiers over Data Streams'' by Tai, et al. and "MISSION: Ultra Large-Scale Feature Selection using Count-Sketches'' by Aghazadeh, et al.).
>
> - In Section 4, we show that just a single JL matrix suffices to find the $\ell_2$ heavy hitters of $w_*$. This was not previously known even in the streaming literature --- that is, even for standard $\ell_2$ heavy hitters.
>
> - In Section 5, we introduce an algorithm for polynomial kernel logistic regression, and logistic regression where the inputs are low-rank tensors. To find the heavy hitters in this setting, it is not possible to directly apply standard streaming techniques without incurring a $d^q$ time complexity --- thus we introduce the result given in Theorem 5.2 (of which the main component is Algorithm 4 in the supplementary material, which we could not include in the main paper due to space constraints). Though this algorithm makes use of the result from the work ``Oblivious Sketching of High-Degree Polynomial Kernels'' by Ahle, et al., Algorithm 4 is not a direct application of that work, and introduces several new ideas. The key new idea in Algorithm 4 is that of iteratively forming prefixes of the $\ell_2$ heavy hitters in $[d]^q$, mode by mode.
>
> Thank you for the comments on the presentation - due to space constraints the writing is a bit dense, but if accepted, we could use the extra page to provide more background on the streaming literature.

---

> > ### Comment · Reviewer_PPCJ · 2021-08-29
> > **thank you**
> >
> > Thank you for clarifying!  There is a bit more to the technique than I originally thought and clearly it is an improvement over the prior work. I will raise my score to 6.

---

### Official Review · Reviewer_pGAj · 2021-07-24

**Rating:** 7
**Confidence:** 4

**Summary:**

The current work deals with linear classification in the streaming, limited space model. The assumption is that the available space is both o(n) and o(d), where n = #data points and d = dimension. The goal is to be able to answer two different types of queries about the weight vector-- either to return a vector that is close to the original weight vector in l2 norm, or to return estimates of any coordinate of the weight vector.

**Ethical Concerns:**

None.

**Limitations And Societal Impact:**

No negative societal impact.

Limitations are discussed above.


**Main Review:**



Previous work has dealt with returning the weight vector estimate where the error is bounded by the L1 norm. The broad structure of the current algorithm is similar -- using a sketch (namely CountSketch) matrix to maintain an estimate of the weight vector, which is updated with the arrival of each point, and then to apply an (existing) recovery procedure in the CountSketch matrix.
There are other results that show that SGD and L-BFGS can be run on sketched data-- while the authors do comment on MISSION, the differences are not entirely clear to me yet. The updates are different, using online gradient descent updates rather than SGD updates, and that reflects in the guarantees, but that does not seem to be the core novelty of the paper.


The main novelty in this work, which allows for the new bounds, is to separate out the two matrices used for sketching -- one is used for storing the weight vector and the other to create a sketch for the incoming points. It is interesting that this simple change (plus its analysis) enables better guarantees. Beyond this observation, most of the techniques are reused from existing work.

The experiments are too brief. The error metric is not exactly the one on which the bound has been proven -- that would be | w-hat_T - w*|_2. However, the authors use a metric that has been used before. Point queries are not empirically evaluated.

There are a number of LateX issues that need to be addressed.
------------
Post rebuttal: Thanks for the answers to the questions. I have upgraded my score.


**Time Spent Reviewing:**

2

---

> ### Author Response · Authors · 2021-08-10
> **Response to Reviewer pGAj**
>
> Thank you very much for your review. We respond below to the concerns you raised.
>
> $\textbf{Comparison with MISSION:}$ The goal of MISSION is to approximate the best $k$-sparse regression solution, while we are concerned with finding the top coordinates of the best (not necessarily sparse) solution, so our results are not directly comparable with MISSION.
>
> One significant difference, between our algorithm and MISSION, is that in MISSION, the weight vector $\beta^{t + 1}$ at iteration $t + 1$ (used to compute the gradient update) is $k$-sparse, and is given by the top $k$ heavy hitters of the contents of Countsketch at iteration $t$. In our algorithm, we do not perform any such truncation, and instead directly use the contents of Countsketch to perform the gradient update.
>
> That feature of our algorithm also allows us to obtain nontrivial results for polynomial kernel logistic regression (as opposed to MISSION, where in the last paragraph of the paper, it is noted that ``the exponential cost is unavoidable with feature extraction.'') In MISSION, the gradient is a scalar multiple of the training examples $X_i$. If $X_i$ is a $q$-th order self-tensoring of a $d$-dimensional point, then feeding it to the Countsketch matrix will already require at least $d^q$ time. In our section on kernel/tensor regression, we show how this $d^q$ time complexity can be completely removed.
>
> Somewhat surprisingly to us, it also seems that MISSION assumes that the design matrix $X$ and the responses $y$ are given by the model $y = X\beta^* + w$, where $\beta^*$ is a $k$-sparse vector and $X, w$ have i.i.d. Gaussian entries (according to Lemmas 2, 3), which is a strong assumption and again makes their results not directly comparable. We note that this is note stated in the introduction of their paper (as far as we can tell), but is critically used in their Lemmas 2 and 3.
>
> ${\bf Technical Novelties:}$ We would like to highlight the additional techniques we introduce in this work, in addition to the observation that the sketches can be separated to obtain the $\ell_2$ guarantee and improve the space complexity. First, in Section 4, we show that with just a single JL matrix (and without any additional sketches used as black boxes) we can obtain the $\ell_2$ point query guarantee. This was not previously known in the streaming literature, even for the standard heavy hitters problem (i.e., without linear classification/regression). Thus, our results advance the streaming literature  beyond the context of classification/regression.
>
> Second, our algorithm mentioned in Theorem 5.2, and given by Algorithm 4 in the supplementary material, is also not known in the streaming literature --- if we are given a sequence of points $x_t \in R^d$ in a streaming, and want to find the $\ell_2$ heavy hitters of $M = \sum_{t = 1}^T x_t^{\otimes q}$, then standard streaming algorithms for $\ell_2$ heavy hitters would require at least $d^q$ time, since they would need to explicitly form the tensor $x_t^{\otimes q}$. Our algorithm is involved: in addition to the recursive tensor sketch of Ahle, et al. (given in the paper ``Oblivious Sketching of High-Degree Polynomial Kernels''), in order to reduce the time complexity from $d^q$ to poly$(dq/\epsilon)$, our algorithm requires the additional ideas of iteratively decoding the heavy coordinates on each mode of the tensor to form the prefixes of the heavy hitters in $[d]^q$. Our fast decoding algorithm for tensors should be of independent interest.
>
> ${\bf Experiments:}$ We were unfortunately not able to include all of our experimental results in the main paper due to space constraints. We include more comprehensive experimental results in Section E of the supplementary material and in the folder Plots-3-Algorithms of the supplementary material. In the revised version, we will include plots of the point query errors (divided by $\|w\|_2$).
>
> An additional example experiment we have is showing that our setting is well-motivated, since the classifier is well-approximated by a sparse vector (this experiment was suggested by Reviewer 3). Here we plot $\frac{\|w_k - w\|_2}{\|w\|_2}$ as a function of $k$, where $w$ is the weight vector obtained by online logistic regression (without any sketching), and $w_k$ is the best $k$-sparse approximation to $w$ (meaning that the nonzero coordinates of $w_k$ are exactly the $k$ largest coordinates of $w$). We obtained the following results when $k = 20,000$, on the Malicious URL dataset (used in our work and in the work ``Sketching Linear Classifiers over Data Streams''):
>
> - When $\lambda = 10^{-3}$, $\frac{\|w_k - w\|_2}{\|w\|_2} < 0.2$.
> - When $\lambda = 10^{-4}$, $\frac{\|w_k - w\|_2}{\|w\|_2}$ is only slightly larger than $0.2$.
> - When $\lambda = 10^{-5}$, $\frac{\|w_k - w\|_2}{\|w\|_2}$ is roughly $0.4$.
>
> Thank you for pointing out the Latex issues - we will address these in the next version.

---

### Author Response · Authors · 2021-08-27
**discussion phase**

Dear reviewers and ACs,

Since the discussion period is about to end soon, we are wondering whether our responses have helped address your questions/concerns or whether we can help further clarify anything.

Thanks again,
The authors

---

### Decision · Program_Chairs · 2021-09-27

**Decision:**

Accept (Poster)

**Comment:**

The reviewers largely agreed that the paper provides a clear improvement over prior work on the problem of learning "heavy-hitter" weights. The reviewers initially debated the value of the problem studied by the authors, and were concerned it might be of limited interest. However, the author response did a good job of further justifying the setting studied.